# Regularization-based Framework for Quantization-, Fault- and Variability-Aware Training

## Abstract

Efficient inference is critical for deploying deep learning models on edge AI devices. Low-bit quantization (e.g., 3- and 4-bit) with fixed-point arithmetic improves efficiency, while low-power memory technologies like analog nonvolatile memory enable further gains. However, these methods introduce non-ideal hardware behavior, including bit faults and device-to-device variability. We propose a regularization-based quantization-aware training (QAT) framework that supports fixed, learnable step-size, and learnable non-uniform quantization, achieving competitive results on CIFAR-10 and ImageNet. Our method also extends to Spiking Neural Networks (SNNs), demonstrating strong performance on 4-bit networks on CIFAR10-DVS and N-Caltech 101. Beyond quantization, our framework enables fault- and variability-aware fine-tuning, mitigating stuck-at faults (fixed weight bits) and device resistance variability. Compared to prior fault-aware training, our approach significantly improves performance recovery under upto 20% bit-fault rate and 40% device-to-device variability. Our results establish a generalizable framework for quantization and robustness-aware training, enhancing efficiency and reliability in low-power, non-ideal hardware.

## 1 Introduction

Deep learning has become ubiquitous in computer vision and broader AI applications, traditionally relying on cloud-based models that transfer data to servers for inference. While effective, this approach suffers from data transfer and power consumption inefficiencies. Edge inference emerges as a compelling alternative, particularly for simple tasks, utilizing low-power accelerators with fixed-point arithmetic and in-memory/near-memory computing architectures (Kukreja et al., 2019; Chen et al., 2019; Chih et al., 2021; Jia et al., 2020; Seo et al., 2022). These architectures, such as crossbar arrays, optimize matrix-vector multiplication through parallel operations, implemented via analog or digital components. Their efficiency stems from **(a)** replacing floating-point operations with fixed-point arithmetic to reduce computational complexity and **(b)** minimizing memory/data transfer through in-memory computation (Figure 2, 1). Further gains can be achieved with emerging technologies such as low-power nonvolatile memory (Deshmukh et al., 2024) and aggressive low-bit quantization, but this introduces a critical trade-off between energy efficiency and model performance (Han et al., 2024; Sun et al., 2023).

Energy-efficient edge devices often encounter circuit non-idealities, such as device-to-device variability in resistance states (Peng et al., 2020; Rasch et al., 2021; Lammie et al., 2022; Deshmukh et al., 2024) and permanent stuck-at (SA) faults (Hanif & Shafique, 2023; Kim et al., 2018), which can severely impact the reliability of neural network inference. In resistive memory-based in-memory computing architectures, resistance states are typically categorized into high-resistance state (HRS) and low-resistance state (LRS). HRS corresponds to a higher resistance value, representing a logical '0,' while LRS corresponds to a lower resistance value, representing a logical '1'; these resistance states can be affected by device-to-device variability (Peng et al., 2020; Rasch et al., 2021; Lammie et al., 2022; Deshmukh et al., 2024) which can degrade inference performance. SA faults occur when weight bits become irreversibly fixed to either the '0' or '1' state, limiting the possible representational range of weights (Hanif & Shafique, 2023; Kim et al., 2018). These issues are inextricably tied to power and area-efficient devices and operating regimes. For example, (Deshmukh et al., 2024) show device-to-device variability of weight bit resistance states increasing as the device

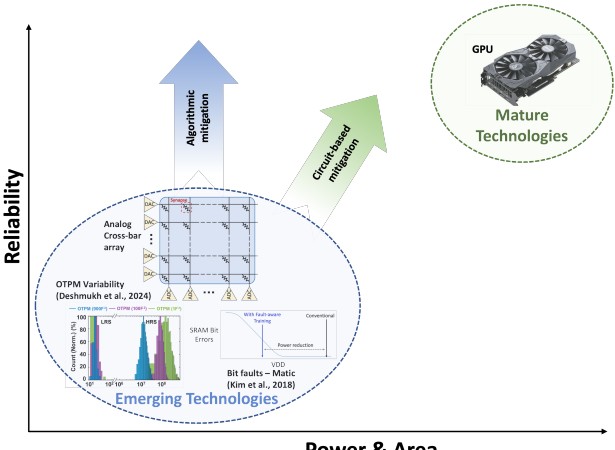

Figure 1: Algorithmic mitigation of emerging technology non-idealities can preserve area and power advantages. One-Time Programmable Memory (OTPM) resistance state (HRS, LRS) variability data taken from Deshmukh et al. (2024) ($F^2$ refers to device scaling, i.e., area), with the bit fault plot from Kim et al. (2018).

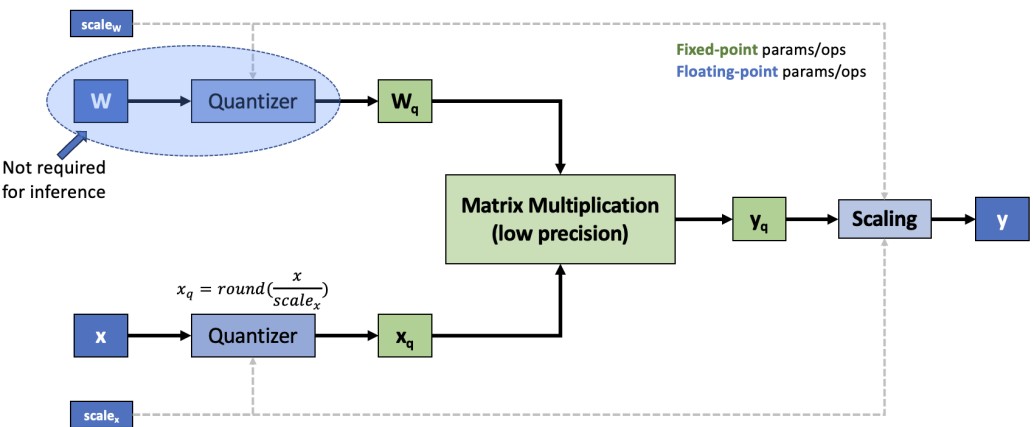

Figure 2: Inference with quantization – Floating-point matrix-vector multiplication is replaced with fixed-point operations, enhancing efficiency. Figure adapted from Esser et al. (2019).

feature size ($F^2$) is reduced and (Kim et al., 2018) show that bit faults are tied to process variance, where device mismatch can create "default" preferred states for bit-cells under low-power read operations (read disturbance) - leading to permanent SA faults. Addressing quantization errors, variability, and hardware faults is therefore, crucial for ensuring robust and reliable inference on edge devices. Figure 1 visually summarizes the challenges facing low-power emerging technology-based inference accelerators and the opportunities presented by algorithmic mitigation approaches.

Low-bit quantization via quantization-aware training (QAT) has been extensively studied, yet existing methods struggle with catastrophic hardware non-differentiability and limited robustness to high fault rates. While custom gradient-based QAT methods (Esser et al., 2019; Choi et al., 2018; Tang et al., 2022) excel on benchmarks like ImageNet(Russakovsky et al., 2015), their reliance on straight-through estimators (STE) limits applicability to hardware-induced discontinuities. Regularization-based approaches (Wess et al., 2018; Solodskikh et al., 2022; Biswas & Ganguly, 2024), though flexible, have not been fully extended to handle variability and extreme fault rates. This paper bridges these gaps by unifying fault-aware and variability-aware training within a regularization framework, enabling robustness upto 20% fault rate and 40% /variability in

low-bit networks. In addition to Artificial Neural Networks (ANNs), we extend our work to enable highly efficient quantized inference in brain-inspired SNNs. Our key contributions can be summarized as follows:

- We introduce a flexible regularization-based QAT approach capable of handling a wide spectrum of quantization schemes, from fixed quantization to learned step sizes and non-uniform learnable quantization.

- We demonstrate our method's effectiveness across multiple networks and datasets, achieving comparable state-of-the-art results for 3- and 4-bit ANNs on CIFAR-10(Krizhevsky et al., 2009) and ImageNet, and for 4-bit SNNs on event-based datasets: CIFAR10-DVS (Li et al., 2017) and N-Caltech 101 (Orchard et al., 2015).

- We introduce fault- and variability-aware training methods that leverage the flexibility of our regularization-based QAT, allowing low-bit models to sustain high performance even under extreme hardware non-idealities. Specifically, our approach demonstrates resilience to up to 20-30% bit-fault rate and 40% device-to-device variability, as validated on CIFAR-10 and ImageNet.

- We present a custom implementation of bit-level multipliers for analog and digital crossbars, optimized for our quantization scheme and seamlessly adaptable to neuromorphic hardware.

## 2 Related Work

### 2.1 Quantization-Aware Training (QAT)

Quantization-aware training methods are broadly categorized by their training methodology and quantization schemes. The two dominant training paradigms are:

- **Custom Gradient-Based Methods**: These approaches, such as (Esser et al., 2019; Choi et al., 2018; Tang et al., 2022; Zhang et al., 2018; Yamamoto, 2021), integrate quantization directly into the neural network computation graph by replacing standard layers with quantized variants. During backpropagation, the non-differentiable quantization operations (e.g., rounding) are handled via gradient approximations, most commonly using straight-through estimators (STE). While these methods achieve state-of-the-art performance on large-scale benchmarks like ImageNet, their reliance on STE limits their ability to handle catastrophic non-differentiability introduced by hardware faults or bit-level variability (Figure 5). For instance, STE approximates gradients for smooth quantization steps but fails to account for abrupt discontinuities caused by stuck-at faults or resistance state fluctuations in memory arrays.

- **Regularization-Based Methods**: Unlike gradient-based approaches, these methods (Wess et al., 2018; Solodskikh et al., 2022; Elthakeb et al., 2019; Biswas & Ganguly, 2024) treat quantization as an implicit constraint during training by adding regularization terms to the loss function. For example, (Wess et al., 2018) uses mean squared error (MSE) between full-precision and quantized weights, while (Solodskikh et al., 2022; Elthakeb et al., 2019) employ sinusoidal regularization to align full-precision weights with discrete quantization levels. A key advantage is their flexibility: since quantization is applied post-training, the network architecture remains unmodified. (Biswas & Ganguly, 2024) demonstrates this by extending regularization to handle stuck-at faults through a Gaussian-like penalty that maps weights to fault-tolerant quantization levels. We build on this foundation by generalizing the regularization framework to address both faults and device variability.

Quantization schemes can also be divided by their numerical representation:

- **Uniform Quantization**: This family, exemplified by (Esser et al., 2019; Choi et al., 2018), employs fixed step sizes (scales) between quantization levels, which can be learned per-layer or per-channel. While simple to implement on hardware (Fig. 2), uniform schemes lack expressiveness for skewed weight distributions, often necessitating higher bit-widths to maintain accuracy.

- **Non-Uniform Quantization**: These methods (Yamamoto, 2021; Jung et al., 2019; Zhang et al., 2018) optimize variable step sizes to better capture the statistical properties of weights and activations. Non-linear approaches like (Yamamoto, 2021; Jung et al., 2019) use companding functions or learned code-books, but require lookup tables (LUTs) during inference, complicating deployment on in-memory computing architectures. In contrast, linear non-uniform schemes like (Zhang et al., 2018) decompose quantized values into bit-weighted sums, enabling efficient crossbar implementations (Figure 8). Our work adopts a similar linear decomposition but integrates it with a regularization-based training objective, allowing simultaneous optimization of bit-multipliers and robustness to hardware imperfections.

## 2.2 Fault and Variability Mitigation

Hardware faults and device variability pose challenges for low-bit quantized models deployed on edge devices:

- **Fault-Aware Training**: Permanent stuck-at (SA) faults, where weight bits are irreversibly stuck at 0/1, are commonly addressed by retraining with fault injection. (Hanif & Shafique, 2023) proposes mapping faulty weights to the nearest valid quantization level during forward passes, while (Kim et al., 2018) uses error masks to simulate bit flips during training. However, these methods focus on 8-bit quantization and small-scale datasets (e.g., MNIST), with limited validation on low-bit networks under moderate-high fault rates (>=10%). Dropout-inspired approaches (Zahid et al., 2020; Koppula et al., 2019) and error correction codes (Reagen et al., 2016) improve tolerance to transient faults but require significant model overcapacity (Koppula et al., 2019), and typically only work for lower fault rates (2.5% with a highly constrained error model in (Zahid et al., 2020), $0.5 - 5\%$ in (Koppula et al., 2019) and 5% in Reagen et al. (2016)) making them impractical for quantized edge models. Our method extends nearest-level mapping with a learnable regularization loss, enabling 4 bit ResNet-18 models to tolerate up to 20% permanent SA faults on ImageNet.

- **Variability-Aware Training**: Device-to-device resistance variability in memory arrays (e.g., $\sigma/\mu$ up to 40% in (Deshmukh et al., 2024)) distorts the effective weights during inference. Traditional mitigation involves chip-in-loop training (Gonugondla et al., 2018; Zhang et al., 2017), where weights are iteratively tuned on physical hardware — a process limited by the lack of batch processing on neuromorphic chips. Simulation frameworks (Peng et al., 2020; Rasch et al., 2021; Lammie et al., 2022) bypass this by profiling variability distributions and injecting noise during training. However, existing implementations treat variability as additive noise, neglecting its dependence on resistance states (HRS/LRS). We explicitly model state-dependent variability during regularization, enabling robust 4-bit networks under 40% $\sigma/\mu$ variability without hardware-in-the-loop iterations. Fig. 3 demonstrates the difference between chip-in-loop training and simulation-based approaches

Our work unifies these directions through a generalized regularization framework, addressing both quantization-aware training and hardware robustness in a single optimization objective. This contrasts with prior efforts that treat quantization, faults, and variability as separate concerns, leading to suboptimal trade-offs between accuracy and reliability.

## 3 Methodology

### 3.1 Preliminaries

Quantization replaces floating-point weights and activations in deep neural networks with low-bit fixed-point representations, reducing memory usage and accelerating computation. An **N**-bit quantization function maps values to one of $2^N$ discrete levels, $l_1, l_2, \ldots, l_{2^N}$, with $2^N - 1$ transition thresholds, $t_1, t_2, \ldots, t_{2^N-1}$, and is defined as follows:

$$Q(x) = \begin{cases} l_1 & \text{if } x < t_1 \\ l_i & \text{if } t_{i-1} \le x < t_i, \quad i = 2, 3, \ldots, 2^N - 1 \\ l_{2^N} & \text{if } x \ge t_{2^N-1} \end{cases} \quad (1)$$

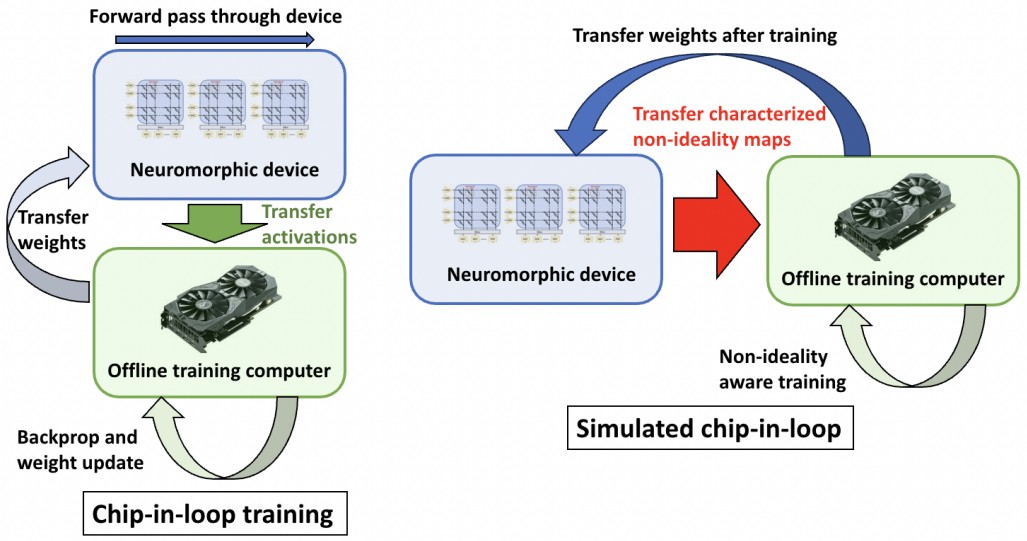

Figure 3: Comparison of chip-in-loop vs simulated chip-in-loop method of hardware-aware optimization. In this work, simulated chip-in-loop model is used to train with device-to-device variability and stuck-at faults (non-idealities to be characterized) awareness

## 3.2 Quantizer Model (N-Multipliers)

In this work, we parameterize the quantization levels as a linear transformation, expressed as:

$$W_{levels} = \{\langle r, S \rangle + c\} \tag{2}$$

where $r$ is the learnable quantization parameter, $S$ denotes a member of the set of complementary bases that, together with $r$, defines the quantization levels, $c$ is the scalar offset of the quantization function and the operator $\langle x, y \rangle$ represents the inner product of $x$ and $y$. For instance, a uniform quantization function with learnable step size (denoted by $r$) is given by:

$$W_{levels} = \left\{ \langle r, S \rangle + c \mid S \in \{0, 1, 2, ..., 2^N - 1\} \right\} \tag{3}$$

Similarly, a linear non-uniform quantization function with learnable bit multipliers (denoted by the N-dimensional vector $r \in \mathbb{R}^N$) is defined as:

$$W_{levels} = \left\{ \langle r, S \rangle + c \mid S \in \{0, 1\}^N \right\} \tag{4}$$

The quantization function maps each full-precision weight to its nearest quantized counterpart:

$$\hat{x} = Q(x, r) = \arg \min_{w_q \in W_{levels}} \mid x - w_q \mid \tag{5}$$

This design enables a flexible quantizer that can be defined to varying levels of freedom - from fixed quantization to learnable step sizes, to linear non-uniform quantization with custom bit multipliers. The N-Multiplier setup allows multiple step sizes, offering hardware efficiency while preserving the structure of N-bit quantization. Although learning all $2^N$ quantization levels would offer maximum flexibility, it would undermine hardware efficiency and the core benefits of N-bit quantization. Figure 4(a) illustrates a sample quantizer function. Comparing a general N-bit quantizer with our formulation, the elements of $W_{levels}$ correspond to the quantization levels $l_1, l_2, \ldots, l_{2^N}$, while the transition thresholds are given by $t_i = \frac{l_i + l_{i+1}}{2}$.

## 3.3 Loss and Learning

We jointly optimize the quantization parameters (learnable step sizes or bit multipliers, depending on the model), offset values, and weights by incorporating an additional quantization-aware loss alongside the

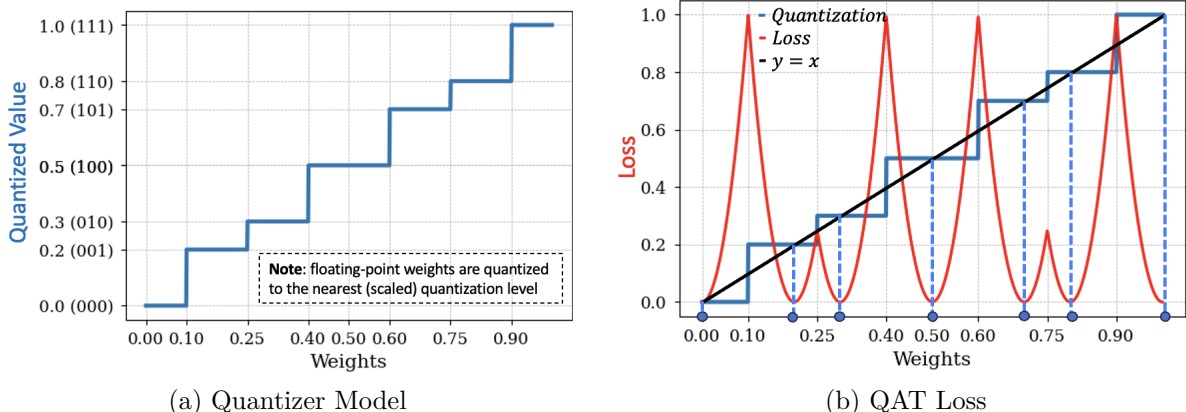

(a) Quantizer Model

(b) QAT Loss

Figure 4: **(a)** Non-uniform quantizer model illustrating the learned bit-multiplier quantization function. **(b)** QAT loss with MSE regularization (shown in red) to minimize quantization error.

standard cross-entropy loss. This enables end-to-end optimization via backpropagation within the standard training pipeline. We have explored two variants of QAT with respect to how model weights are handled during the forward pass in the training stage: In the first variant, the full-precision weights remain unchanged during the forward pass. In the second variant, the weights are quantized and appropriately scaled during the forward pass using a quantization function with an overloaded *straight-through estimator* (**STE**) in the backward pass. We find that the second variant yields better QAT outcomes. During training, weights remain in full precision but progressively align with their quantized counterparts under the influence of the quantization-aware loss. Post-training, full-precision weights are mapped to their nearest quantized values for efficient inference.

### 3.4 Quantization-Aware Training Loss

We define a regularization loss to minimize the squared error between each weight and its nearest quantized value. To ensure balanced gradient contributions across layers, we introduce a layer-specific scaling factor. The total loss is formulated as:

$$\mathcal{L} = \mathcal{L}_{CE} + \lambda \sum_{l=1}^{L} \alpha_l \sum_{i=1}^{n_l} \min_{w_q \in W_{levels}^l} | w_i - w_q |^2 \tag{6}$$

where $\mathcal{L}_{CE}$ is the cross-entropy loss, $W_{levels}^l$ is the set of quantized weights for layer $l$, defined by parameters $r^l$ and $c^l$, $L$ is the total number of layers, and $n_l$ is the number of trainable weights in layer $l$. The term $\alpha_l$ is a layer-wise scaling factor, and $\lambda$ controls the regularization strength. Following (Esser et al., 2019), we set $\alpha_l$ as $1/\sqrt{N \cdot Q_P}$, where $Q_P$ is $2^b - 1$ for activations (unsigned data) and $2^{b-1} - 1$ for weights (signed data), with $b$ denoting the number of bits. Figure 4(b) illustrates the regularization loss for a sample weight under the linear non-uniform quantization model with an arbitrary bit multipliers vector ($r$). Equivalently, the loss can be expressed as a function of the weights and quantization parameters. This formulation jointly optimizes the overall objective and the quantization parameters defining the quantization function itself:

$$\mathcal{L} = \mathcal{L}_{CE} + \lambda \sum_{l=1}^{L} \alpha_l \sum_{i=1}^{n_l} | w_i - Q(w_i, r^l) |^2 \tag{7}$$

### 3.5 SNN Training

Spiking Neural Networks (SNNs) are biologically inspired models that process information using discrete spike events, making them inherently more energy-efficient compared to traditional artificial neural networks (ANNs). Due to their event-driven computation and sparse activation patterns, SNNs are particularly well-suited for low-power neuromorphic hardware (Bouvier et al., 2019). Given their emphasis on efficiency,

we found it natural to extend our quantization-aware training method to SNNs, aiming to achieve highly efficient, low-bit quantized inference while maintaining competitive performance.

SNNs inherently produce *quantized activations* in the form of *spike trains*, we thus need to solely quantize the weights of the network. We use a Leaky Integrate-and-Fire (LIF) model (Gerstner & Kistler, 2002) for the spiking neuron in our SNN models. These discrete-time equations describe its dynamics:

$$H[t] = V[t-1] + \beta(X[t] - (V[t-1] - V_{reset})) \tag{8}$$
$$S[t] = \Theta(H[t] - V_{th}) \tag{9}$$
$$V[t] = H[t] \ (1 - S[t]) + V_{reset} \ S[t] \tag{10}$$

where $X[t]$ denotes the input current at time step $t$. $H[t]$ denotes the membrane potential following neural dynamics and $V[t]$ denotes the membrane potential after a spike at step $t$, respectively. The model uses a firing threshold $V_{th}$ and utilizes the Heaviside step function $\Theta(x)$ to determine spike generation. The output spike at step $t$ is denoted by $S[t]$, while $V_{reset}$ represents the reset potential following a spike. The membrane decay constant is denoted by $\beta$. To facilitate error backpropagation, we use the surrogate gradient method (Neftci et al., 2019), defining $\Theta'(x) = \sigma'(x)$, where $\sigma(x)$ is the arctan surrogate function (Fang et al., 2021). The remaining part of the training/quantization follows that of the non-spiking networks described earlier.

### 3.6 Fault-Aware Training

We propose a two-pronged approach to mitigate SA faults in quantized neural networks. First, we emulate *nearest valid level mapping* techniques used in fault-aware training (Hanif & Shafique, 2023) by periodically (every 4 epochs) replacing fault-affected weights in the model with their nearest valid quantization levels. Second, we introduce a fault-aware modification to our regularization loss, designed to prevent weight configurations that become unattainable due to SA faults. Following (Biswas & Ganguly, 2024), we incorporate a *validity* term that constrains weights to achievable quantization levels while excluding those rendered unreachable by faulty bits. The *validity* term is defined per layer as a binary mask that indicates whether a given weight can attain a particular quantization level (1 if achievable, 0 otherwise). This modification updates the quantization-aware training loss from Equation 6 as follows:

$$\mathcal{L} = \mathcal{L}_{CE} + \lambda \sum_{l=1}^{L} \alpha_l \sum_{i=1}^{n_l} \min_{w_q \in W^l_{levels}} (val^l_{i,q} \mid w_i - w_q \mid^2 + (1 - val^l_{i,q}) \cdot \Delta) \tag{11}$$

where $val^l_{i,q}$ represents the *validity* term for weight $w_i$ in layer $l$ with respect to the quantization level $w_q \in W^l_{levels}$. If $w_i$ can reach $w_q$, then $val^l_{i,q} = 1$; otherwise, $val^l_{i,q} = 0$. The term $\Delta$ is a large constant that penalizes unreachable quantization levels, effectively excluding them from optimization. Figure 5 (a) illustrates the impact of stuck-at faults on quantization states and how the fault-aware loss formulation accounts for them. For fault-aware retraining and fine-tuning experiments, we initialize with models pretrained using QAT and subsequently apply fault-aware training for a limited number of epochs.

### 3.7 Variability-Aware Training

For variability-aware training, we note that the concept of bit-multipliers can be extended to represent device-to-device LRS variability by simply multiplying the $r$ vector with the characterized variability map. The variability-aware quantizer definition equation then becomes:

$$W^l_{levels} = \left\{ \langle var^l \odot r^l, S \rangle + c \mid S \in \{0,1\}^N \right\} \tag{12}$$

where $var^l$ represents the characterized device-to-device LRS variability map of the weight-bit cells in layer $l$, and $r^l$ denotes the ideal learned bit-multipliers for layer $l$. In this work, we primarily focus on LRS variability, as it significantly impacts the deviation between trained weights and their on-device implementations. However, HRS variability can also be incorporated within the same framework by defining a parallel set of "0" bit-multipliers and perturbing them with the characterized HRS variability. Figure 5(b) illustrates the impact of LRS variability on quantization states and how the regularization function adapts to mitigate it.

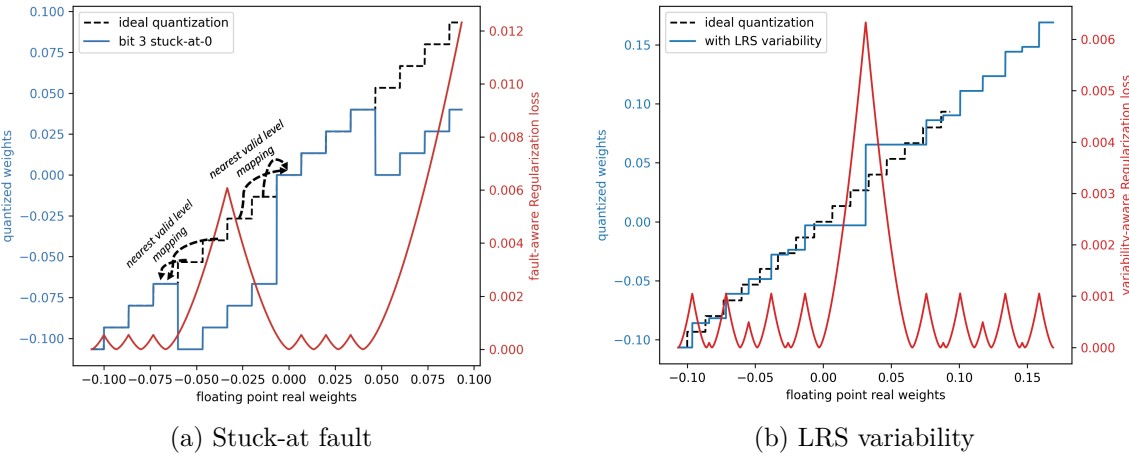

(a) Stuck-at fault        (b) LRS variability

Figure 5: **(a)** Example of a quantized 4-bit weight with a stuck-at-0 fault (third bit stuck at 0 in 4-bit quantization) and its corresponding fault-aware loss. **(b)** Example of a quantized 4-bit weight affected by LRS variability and its corresponding variability-aware loss. **Note**: For clarity in visualization, uniform quantization is used as the base/ideal quantization.

## 4 Experiments

We initialize quantized networks using weights from a pre-trained full-precision model with the same architecture, then fine-tune within the quantized space, which has been shown to enhance performance (McKinstry et al., 2018; Sung et al., 2015; Mishra & Marr, 2017). We quantize input activations and weights to 3- or 4-bits for all matrix multiplication layers, except for the first and last layers. This approach, commonly employed for quantizing deep networks, has demonstrated effectiveness with minimal overhead (Esser et al., 2019). Both weights and quantization parameters (bit multipliers and offset values) are trained using SGD with a momentum of 0.9 and a cosine learning rate decay schedule (Loshchilov & Hutter, 2016). For effective QAT, we apply a schedule to the regularization hyperparameter $\lambda$, starting with a low value, maintaining it for most of the training epochs, and then exponentially increasing $\lambda$ in the final 20 epochs.

The exact range of the $\lambda$ schedule depends on the variant of QAT being implemented. For the *first variant*, where the full-precision weights are left unchanged during the forward pass, we start training with $\lambda = 100$ and increase it to a final value of $\lambda = 2000$ over the last 20 epochs. The learning rate for the quantization model parameters ($r$) is set to $10^{-6}$. For the *second variant*, where the weights are quantized during the forward pass using a quantization function with a straight-through estimator (STE) in the backward pass, we modify the $\lambda$ schedule to start at 0.1 and increase to 1.0 over the final 20 epochs. In this case, the quantization model parameters ($r$) are trained with a learning rate of 0.001. The $\lambda$ values in the *second variant* are scaled down because quantization awareness is already incorporated through the forward pass. At the same time, the learning rate for the quantization model parameters is proportionally scaled up to maintain comparable quantization model learning. For Fault-Aware and Variability-Aware Training, only the *first variant* is implemented, as these tasks are heavily dependent on the regularization loss. The use of STE is less meaningful in this context due to the arbitrary nature of quantization functions under faulty and variability-afflicted bit conditions. Finally, since our regularization-based approach is strictly applicable to the weights, we use LSQ (Esser et al., 2019) to handle activation quantization.

### 4.1 ANN Training

We use the ResNet-18 and ResNet-50 architectures (He et al., 2016) for experiments on the CIFAR-10 (Krizhevsky et al., 2009) and ImageNet (Russakovsky et al., 2015) datasets. First the full-precision (fp32) base models are trained. Full precision models are trained for 200 epochs on CIFAR-10 and 90 epochs on ImageNet, with a learning rate of 0.01. Then, for Quantization-Aware Training, we train the quantized models for the same number of epochs, with the same learing rate (0.01) for the weights, using the pre-

trained full-precision (fp32) models as the staring point and $\lambda$ schedule and quantization model parameters ($r$) learning rate as described above, depending on the QAT variant. For ImageNet, we preprocess images by resizing them to $256 \times 256$ pixels. During training, random $224 \times 224$ crops and horizontal flips are applied with a probability of 0.5. At inference, a center crop of $224 \times 224$ is used. For CIFAR-10, the training data is augmented by padding images with 4 pixels on each side, followed by random $32 \times 32$ crops, and horizontal flips are applied with a probability of 0.5.

## 4.2 SNN Training

We use the ResNet-19 (Zheng et al., 2021) and VGG-11 (Simonyan & Zisserman, 2014) models, adapting them to SNNs by replacing all ReLU activation functions with LIF modules and substituting max-pooling layers with average pooling operations. The baseline training method follows the implementation and data augmentation technique used in NDA (Li et al., 2022). The weights and other parameters are trained with learning rates of 0.01 and 0.001, respectively. We evaluate on the CIFAR10-DVS (Li et al., 2017) and N-Caltech 101 (Orchard et al., 2015) benchmarks. N-Caltech 101 consists of 8,831 DVS images converted from the original Caltech 101 dataset, while CIFAR10-DVS comprises 10,000 DVS images derived from the original CIFAR10 dataset. For both datasets, we apply a 9:1 train-validation split and resize all images to $48 \times 48$. Each sample is temporally integrated into 10 frames using SpikingJelly (Fang et al., 2023). We set $V_{reset} = 0$ and the membrane decay $\beta = 0.25$.

## 4.3 Fault-Aware Training

We evaluate our Fault-Aware Training method primarily in two ways: full fault-aware retraining and few-epoch fault-aware fine-tuning. For a smaller benchmark like CIFAR-10, we perform fault-aware retraining for the same number of epochs as the original QAT. In contrast, for a larger benchmark like ImageNet, we employ fault-aware fine-tuning on trained quantized models for 20 epochs, applying the $\lambda$ scaling schedule in the final 10 epochs. Our experiments analyze various levels of stuck-at (SA) fault density. Figure 6 demonstrates the effectiveness of our approach on the CIFAR-10 benchmark using a VGG-13 architecture, while Figure 7(a) presents results on ImageNet with a ResNet-18 architecture. In both cases, we compare our method against the *nearest valid level* mapping approach (Hanif & Shafique, 2023; Kim et al., 2018), applied to 3-bit and 4-bit quantization for mitigating permanent stuck-at faults, and show substantial improvements.

## 4.4 Variability-Aware Training

We evaluate our Variability-Aware Training using a setup similar to that of Fault-Aware Training on the ImageNet benchmark, performing 20 epochs of Variability-Aware fine-tuning on trained quantized models. Our experiments consider $\sigma/\mu$ values ranging from 0.05 to 0.4 for the weight-bit low-resistance states, representing characterized device-to-device variability. We find our fine-tuning approach to be highly robust, even under severe variability conditions. Figure 7(b) illustrates the effectiveness of our method on the ImageNet benchmark using a ResNet-18 architecture. We compare our results against the simulated *chip-in-loop* approach (Gonugondla et al., 2018; Peng et al., 2020; Rasch et al., 2021), applied to 4-bit quantization for mitigating device-to-device variability in weight bits, and demonstrate significant improvements.

# 5 Results and Analysis

## 5.1 Performance of Quantized ANNs and SNNs

We compare our quantized ANN experiments against several conventional baselines and method variations. In Tables 1 and 2, we present different variations of our approach, distinguished by the "Quantization Type". "Fixed levels" refers to a setting where quantization parameters remain unchanged and are not learned. "Learned scale" corresponds to learning a uniform step size, while "Learnable non-uniform" represents our final method, where both the weights and the non-uniform quantizer parameters are jointly optimized.

Tables 1 and 2 present our quantized ANN results for CIFAR-10 and ImageNet, respectively. Among our method variants, we observe that jointly learning non-uniform quantizer parameters alongside the weights

Table 1: Accuracy (%) for 3- and 4-bit quantized ResNet-18 models on CIFAR-10. FP denotes full-precision (32-bit) accuracy, $\Delta$ FP denotes accuracy difference compared to the corresponding FP network. Best relative performances for each bit-width are marked in **bold**. Baseline results taken from respective works.

| Method | Quantization Type | FP | W4/A4 ($\Delta$ FP) | W3/A3 ($\Delta$ FP) |
|---|---|---|---|---|
| L1 Reg (Alizadeh et al., 2020) | No QAT | 93.54 | 89.98 ($-3.56$) | - |
| BASQ (Kim et al., 2022) | Binary Search | 91.7 | 90.21 ($-1.49$) | - |
| LTS (Zhong et al., 2022) | Lottery | 91.56 | 91.7 ($+0.1$) | 90.58 ($-0.98$) |
| PACT (Choi et al., 2018) | Learned scale | 91.7 | 91.3 ($-0.4$) | 91.1 ($-0.6$) |
| LQ-Nets (Zhang et al., 2018) | Linear non-uniform | 92.1 | - | 91.6 ($-0.5$) |
| LCQ (Yamamoto, 2021) | Non-linear | 93.4 | 93.2 ($-0.2$) | 92.8 ($-0.6$) |
| Ours | Fixed levels | 93.26 | 92.23 ($-1.03$) | 91.68 ($-1.58$) |
| Ours (N-Multipliers) | Learnable non-uniform | 93.26 | **93.50 ($+0.24$)** | **92.84 ($-0.42$)** |

Table 2: Accuracy (%) for 4-bit and 3-bit quantized ResNet-18 models on ImageNet. FP denotes full-precision (32-bit) accuracy, $\Delta$ FP denotes accuracy difference compared to the corresponding FP network. Best relative performances are marked in **bold**. Baseline results taken from respective works. (KD=Knowledge Distillation)

| Method | Quantization Type | FP | W4/A4($\Delta$ FP) | W3/A3($\Delta$ FP) |
|---|---|---|---|---|
| L1 Reg (Alizadeh et al., 2020) | No QAT | 69.7 | 57.5 ($-12.5$) | - |
| SinReQ (Elthakeb et al., 2019) | Sine reg. | 70.5 | 64.6 ($-5.9$) | 61.95 ($-8.55$) |
| LTS (Zhong et al., 2022) | Lottery | 69.6 | 68.3 ($-1.3$) | 66.3 ($-3.3$) |
| PACT (Choi et al., 2018) | Learned scale | 70.2 | 69.2 ($-1$) | 68.1 ($-2.1$) |
| LSQ (Esser et al., 2019) | Learned scale + KD | 70.5 | 71.1 ($+0.6$) | 70.2 ($-0.3$) |
| LQ-Nets (Zhang et al., 2018) | Linear non-uniform | 70.3 | 69.3 ($-1.0$) | 68.2 ($-2.1$) |
| QIL (Jung et al., 2019) | Non-linear | 70.2 | 70.1 ($-0.1$) | 69.2 ($-1$) |
| QSin (Solodskikh et al., 2022) | Sine reg. | 69.8 | 69.7 ($-0.1$) | - |
| LCQ (Yamamoto, 2021) | Non-linear | 70.4 | 71.5 ($+$**1.1**) | 70.6 ($+$**0.2**) |
| Ours | Fixed levels | 69.6 | 68.2 ($-1.4$) | - |
| Ours | Learned scale | 69.6 | 69.4 ($-0.2$) | - |
| Ours (N-Multipliers) | Learnable non-uniform | 69.6 | 69.6 ($-0.0$) | 68.08 ($-1.5$) |
| Ours (N-Multipliers+STE) | Learnable non-uniform | 69.6 | 70.27 ($+0.67$) | 69.1 ($-0.5$) |

Table 3: Accuracy (%) for 4-bit and 3-bit quantized ResNet-50 models on ImageNet. FP denotes full-precision (32-bit) accuracy, $\Delta$ FP denotes accuracy difference compared to the corresponding FP network. Best relative performances are marked in **bold**. Baseline results taken from respective works. (KD=Knowledge Distillation)

| Method | Quantization Type | FP | W4/A4($\Delta$ FP) | W3/A3($\Delta$ FP) |
|---|---|---|---|---|
| LTS (Zhong et al., 2022) | Lottery | 76.15 | 74.19 ($-1.96$) | 72.86 ($-3.3$) |
| PACT (Choi et al., 2018) | Learned scale | 76.9 | 76.5 ($-0.4$) | 75.3 ($-1.6$) |
| LSQ (Esser et al., 2019) | Learned scale + KD | 76.9 | 76.7 ($-0.2$) | 75.8 ($-1.3$) |
| LQ-Nets (Zhang et al., 2018) | Linear non-uniform | 76.4 | 75.1 ($-1.3$) | 74.2 ($-2.2$) |
| LCQ (Yamamoto, 2021) | Non-linear | 76.8 | 76.6 ($-0.2$) | 76.3 ($-0.5$) |
| Ours (N-Multipliers) | Learnable non-uniform | 75.5 | 75.6 ($+0.1$) | 74.38 ($-1.12$) |
| Ours (N-Multipliers + STE) | Learnable non-uniform | 75.5 | 75.81 ($+$**0.31**) | 75.12 ($-$**0.4**) |

yields the best performance. When compared to other methods, our approach matches or outperforms existing techniques, with 4-bit ResNet-18 (W4/A4, denoting 4-bit weights and 4-bit activations) achieving

a **0.24%** accuracy improvement over full-precision (FP) on CIFAR-10 and matching FP performance on ImageNet. For 4-bit quantized SNNs (Table 4), we observe performance gains on N-Caltech 101 and slight accuracy drops on CIFAR10-DVS compared to FP. Notably, occasional performance improvements in both 4-bit ANNs and SNNs can be attributed to the regularization effect induced by our quantization loss.

Table 4: Accuracy (%) for 4-bit quantized SNNs on CIFAR10-DVS and N-Caltech 101. FP denotes full-precision (32-bit) accuracy, $\Delta$ FP denotes accuracy difference compared to the corresponding FP network.

| Dataset | Model | FP | W4 ($\Delta$ FP) |
|---------|-------|-----|------------------|
| CIFAR10-DVS | Spiking VGG-11 | 71.92 | 71.84 ($-0.08$) |
| CIFAR10-DVS | Spiking ResNet-19 | 72.91 | 72.14 ($-0.77$) |
| N-Caltech 101 | Spiking VGG-11 | 73.19 | 74.18 ($+0.99$) |
| N-Caltech 101 | Spiking ResNet-19 | 75.27 | 75.93 ($+0.66$) |

## 5.2 Robustness to Faults

SA faults represent severe hardware non-idealities, where each faulty bit effectively halves the range of possible weight values. Our approach, which combines *nearest valid level* mapping with a fault-aware regularization loss, demonstrates strong robustness even under high SA fault densities and low-bit quantization, as shown in Figure 6 and Figure 7(a).

The approaches to fault-aware training for permanent bit faults can be divided into four distinct levels: *basic awareness* (i.e., periodic remapping of network weights based on fault maps—for example, every 4 epochs), *passive mitigation* (i.e., nearest valid level mapping, as in Hanif & Shafique (2023), where errors due to bit faults are minimized during the mapping of floating-point weights to fixed-point values), *active mitigation* (as in the method proposed by Biswas & Ganguly (2024), which uses regularization-based fault-aware training without nearest valid level mapping), and *active+passive mitigation* (this work, which combines regularization-based fault-aware training with nearest valid level mapping).

We perform fault-aware training on the ResNet-18 network using ImageNet under different bit-fault levels for all four configurations described above. As shown in Figure 7(a), *active+passive mitigation* performs best in preserving network performance in the presence of bit faults, followed by *active mitigation*, which outperforms *passive mitigation*, while *basic awareness* performs the worst. We consistently outperform the established *nearest valid level mapping* approach across the entire range of bit-fault rates.

Specifically, our method maintains strong resilience to SA faults, tolerating up to 20% and 30% bit-fault rate in 4-bit and 3-bit VGG-13 models on CIFAR-10, respectively, and up to 20% in the 4-bit ResNet-18 and ResNet-50 models on ImageNet. Table 5 compares our fault-aware training results with the current literature on fault-aware training - FAQ (Hanif & Shafique, 2023) and Matic Kim et al. (2018) retrain with *nearest valid level mapping*, EDEN (Koppula et al., 2019) performs dropout-inspired generic fault-aware training, and QFALT (Biswas & Ganguly, 2024) uses a Regularization-based method. It is important to note that EDEN (Koppula et al., 2019) reports a maximum tolerable bit fault rate for which test accuracy degradation remains below 1%. Furthermore, in all fault-aware training experiments described in the paper, test accuracy inevitably collapses to approximately 0% as the bit fault rate approaches 10%.

## 5.3 Robustness to Weight-bit Variability

Device-to-device variability in weight-bit resistance states is a significant challenge in emerging in-memory computing platforms (Deshmukh et al., 2024; Peng et al., 2020; Rasch et al., 2021). Our approach enhances the *simulated chip-in-loop* method by introducing a variability-aware regularization loss, ensuring robust performance under high levels of weight-bit LRS variability (up to 40% $\sigma/\mu$ in device-to-device variability) and low-bit quantization, as shown in Figure 7(b). Table 6 summarizes comparisons with the current literature on variability-aware training, specifically NeuroSim Peng et al. (2020) and CoMN Han et al. (2024), which use the *simulated chip-in-loop* method.

Table 5: Comparison of fault-aware training approaches. $\Delta$ FP denotes the accuracy difference compared to the corresponding FP32 baseline network.

| Method | Model | Dataset | Precision | Bit-Fault | Accuracy (%) ($\Delta$ FP) |
|--------|-------|---------|-----------|-----------|----------------------------|
| FAQ | ResNet-18 | CIFAR-10 | 8-bit | 20% | 90.4 ($-2.94$) |
| QFALT | 10-layer CNN | CIFAR-10 | 3-bit | 10% | 88.8 ($-2$) |
| EDEN | ResNet-101 | CIFAR-10 | 8-bit | 4% | 92.14 ($-1$) |
| EDEN | DenseNet201 | ImageNet | 8-bit | 1.5% | 73.5 ($-1$) |
| Ours | VGG-13 | CIFAR-10 | 3-bit | 10% | 92.3 ($-0.7$) |
| Ours | VGG-13 | CIFAR-10 | 3-bit | 20% | 92 ($-1$) |
| Ours | ResNet-18 | CIFAR-10 | 3-bit | 10% | 92.86 ($-0.4$) |
| Ours | ResNet-18 | CIFAR-10 | 3-bit | 20% | 92.79 ($-0.47$) |
| Ours | ResNet-18 | ImageNet | 4-bit | 10% | 67.34 ($-2.3$) |
| Ours | ResNet-50 | ImageNet | 4-bit | 5% | 74.08 ($-1.4$) |
| Ours | ResNet-50 | ImageNet | 4-bit | 10% | 73.64 ($-1.85$) |

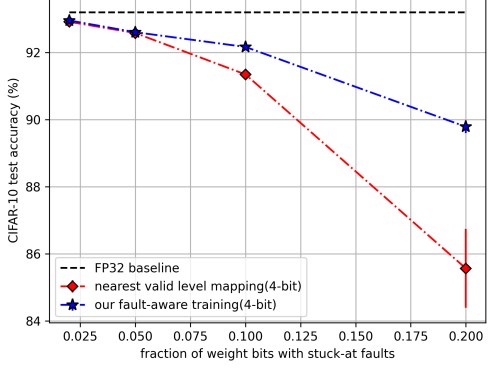

(a) 4-bit VGG-13 model with SA faults

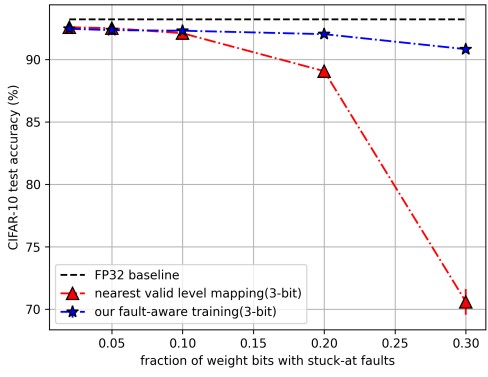

(b) 3-bit VGG-13 model with SA faults

Figure 6: Fault-Aware Training evaluated on CIFAR-10 using 3- and 4-bit ResNet-18 models.

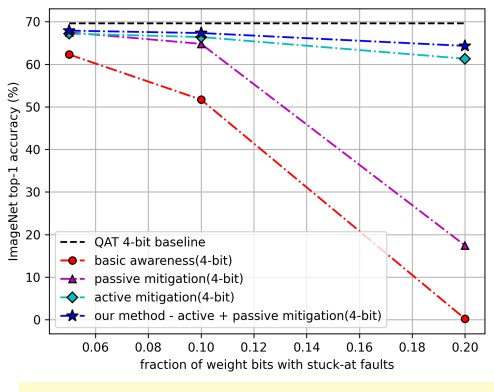

(a) 4-bit ResNet-18 model with SA faults

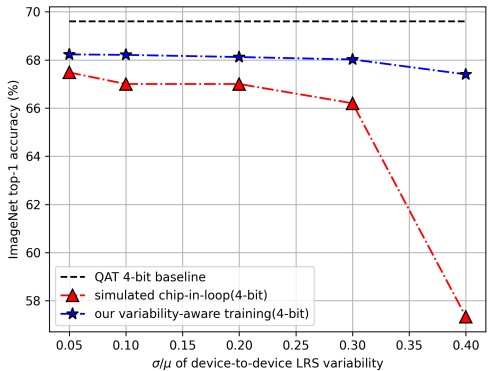

(b) 4-bit ResNet-18 model with weight-bit variability

Figure 7: Fault- and Variability-Aware Training evaluated on ImageNet using 4-bit ResNet-18 model.

Table 6: Comparison of variability-aware training approaches. $\Delta$ FP denotes the accuracy difference compared to the corresponding FP32 baseline network.

| Method | Model | Dataset | Precision | Var. $(\sigma/\mu)$ | Accuracy (%) ($\Delta$ FP) |
|---|---|---|---|---|---|
| NeuroSim | VGG-8 | CIFAR-10 | 6-bit | 2% | 85 $(-3)$ |
| CoMN | ResNet-50 | ImageNet | 8-bit | 3% | 73.1 $(-1.9)$ |
| Ours | VGG-13 | CIFAR-10 | 4-bit | 20% | 92.35 $(-0.65)$ |
| Ours | ResNet-18 | ImageNet | 4-bit | 20% | 68.12 $(-1.5)$ |
| Ours | ResNet-18 | ImageNet | 4-bit | 40% | 67.4 $(-2.2)$ |
| Ours | ResNet-50 | ImageNet | 4-bit | 20% | 73.42 $(-2.1)$ |
| Ours | ResNet-50 | ImageNet | 6-bit | 20% | 74.90 $(-0.6)$ |

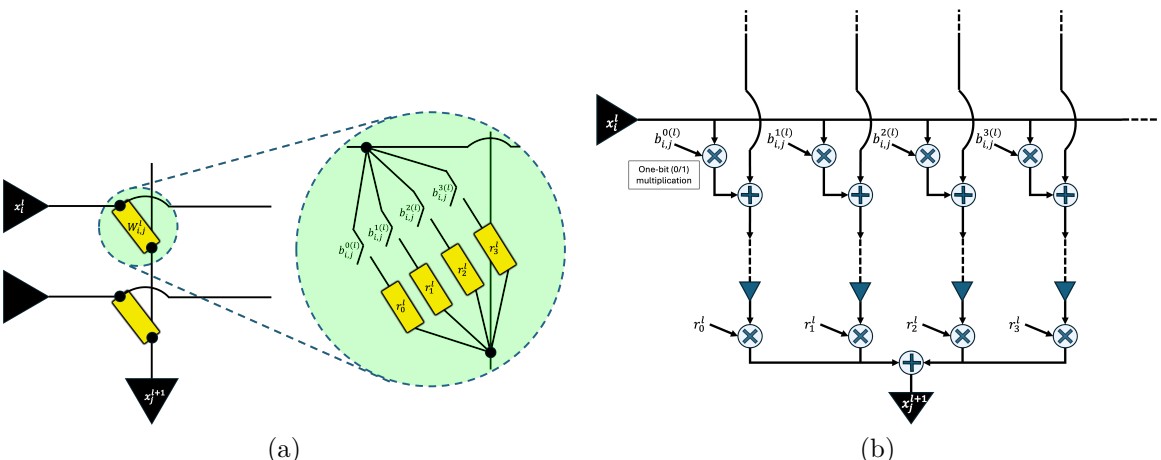

|              (a)                           (b)              |

Figure 8: Custom bit multipliers implementation: **(a)** Analog implementation. **(b)** Digital implementation. While uniform quantization employs bit multipliers $(r_0^l, r_1^l, r_2^l, r_3^l)$ in power-of-2 proportions $(1, 2, 4, 8)$, we propose learning custom multiplier factors instead.

## 5.4 Hardware Compatibility

Figure 8 illustrates the implementation of custom bit multipliers in both analog and digital crossbar arrays. In analog arrays, this implementation incurs no additional cost, as it only requires adjusting the bit-multiplier conductance values from power-of-2 proportions to custom values. In digital arrays, the multiply-accumulate operation remains entirely fixed-point, with the custom bit-multiplier scaling absorbed into the final floating-point scaling operation, which is a common component of quantization schemes (Esser et al., 2019) (see Fig. 2). This approach effectively minimizes the overhead associated with floating-point bit multipliers. Learning custom bit multipliers within QAT enables highly efficient low-bit quantization models that seamlessly integrate with standard in-memory computing architectures.

## 5.5 Challenges, Limitations and Scope

In this paper, we primarily focus on Quantization-Aware Training (QAT) and related methods to perform Fault-Aware and Variability-Aware training on top of low-bit quantized models, in order to mitigate characterized hardware non-idealities. As we have shown, this enables the development of low-bit quantized models that are robust to high levels of stuck-at faults and LRS variability. However, as a QAT-based approach, our method requires extensive training to converge on the optimal configuration of model weights and quantization parameters for low-bit quantization (specifically 3-bit and 4-bit, as explored in this work). This typically demands the same number of training epochs as the original full-precision training. Furthermore, even the Fault-Aware and Variability-Aware fine-tuning on top of the quantized model requires up to 20 additional

training epochs. While this training requirement is manageable for standard convolutional networks (such as the ResNet family) on computer vision benchmarks, it poses a significant challenge for transformer-based architectures Vaswani et al. (2017), including Vision Transformers Dosovitskiy et al. (2020) and language models Radford et al. (2019). These models require extensive pre-training on large-scale unlabeled datasets, making it impractical to apply QAT-based methods in such settings.

As a result, the focus of model quantization research for language models and transformer-based architectures has shifted toward Post-Training Quantization (PTQ) techniques, which typically develop separate strategies for weight quantization and key-value (KV) cache quantization. For weight quantization, the quantization error in the linear layer outputs is usually minimized over a small calibration dataset using linear algebra techniques to determine optimal quantization parameters Cheng et al. (2023); Frantar et al. (2022); Li et al. (2025). Alternatively, low-rank adapters can be trained for task-specific fine-tuning on top of quantized base language models Dettmers et al. (2023), or low-rank approximations of the eigen decomposition of quantization errors can be used to enhance quantized linear layer outputs Liu et al. (2024). Statistical techniques are also applied to enable mixed-precision quantization Dettmers et al. (2022). For KV-cache quantization, statistical methods are often used to perform token-specific or channel-specific quantization and to identify important features for mixed-precision representation Tao et al. (2025).

Quantization-error minimization PTQ techniques appear to be a natural extension of regularization-based QAT methods, with some important distinctions: **first**, these techniques minimize the error in the quantized linear layer outputs rather than the weight quantization error; and **second**, the underlying weights are kept static, with only the quantization parameters being optimized. We believe there is potential to incorporate linear non-uniform quantization through our N-multipliers formulation into the PTQ workflow for transformer-based models.

## 6 Conclusions and Future Work

In this paper, we introduce a flexible regularization-based framework for quantization-aware training (QAT) that generalizes across various quantization schemes, ranging from fixed quantization to learned step sizes and learned bit multipliers for linear non-uniform quantization. Our approach achieves performance comparable to state-of-the-art QAT methods, as demonstrated through benchmarking on CIFAR-10 and ImageNet using ResNet-18 and ResNet-50 networks. Furthermore, we extend our framework to fault- and variability-aware training, effectively mitigating the impact of permanent stuck-at faults and device-to-device variability in weight bits—key challenges in low-power and in-memory computing-based neural network accelerators. Our method outperforms standard mitigation techniques in these scenarios. We apply our method to spiking neural networks (SNNs) using spiking variants of VGG-11 and ResNet-19 trained on standard neuromorphic benchmarks (CIFAR10-DVS and N-Caltech 101), achieving 4-bit quantization while preserving floating-point baseline performance.

Future research directions include constraining learned step sizes and bit multipliers (for both weights and activations) to enable low-bit, purely fixed-point inference without requiring inter-layer floating-point scaling and developing effective PTQ techniques for performing linear non-uniform quantization of very large models. The advancements proposed in this paper seek to improve the efficiency and robustness of quantized neural networks, particularly for resource-constrained environments and hardware non-idealities Furthermore, we aim to extend our method to architectures such as Large Language Models (LLMs) and Vision Transformers (ViTs), broadening its applicability.

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
