# OpenReview forum: "Regularization-based Framework for Quantization-, Fault- and Variability-Aware Training"
_TMLR — Rejected by TMLR_

### Review · Reviewer_91Ln · 2025-05-22

**Summary Of Contributions:**

The paper proposes a unified, regularization-based framework for Quantization-Aware Training (QAT) that effectively addresses low-bit quantization, hardware-induced bit faults, and device-to-device variability. Key contributions include:

* A flexible QAT approach supporting fixed, learnable step-size, and learnable non-uniform quantization via bit multipliers.
* State-of-the-art performance for 3- and 4-bit quantized Artificial Neural Networks (ANNs) on CIFAR-10 and ImageNet, and for 4-bit Spiking Neural Networks (SNNs) on CIFAR10-DVS and N-Caltech 101.
* Novel fault- and variability-aware training strategies that improve robustness under up to 20–30% stuck-at fault rates and 40% device variability, outperforming prior methods.
* Hardware-compatible implementation of custom bit-multipliers for analog and digital crossbars, enabling efficient deployment.

This work offers a generalizable and practical solution for robust, energy-efficient neural network inference on emerging non-ideal edge hardware.

**Audience:**

Yes

**Claims And Evidence:**

Yes

**Requested Changes:**

Critical Changes:

1. Clarify Limitations Regarding Post-Training Quantization:
   The paper should explicitly acknowledge that the method does not support post-training quantization and explain why this limitation exists. A brief discussion of potential future solutions would also be helpful.

2. Ablation Study on Loss Components:
   Include ablation experiments to isolate the effects of different components in the loss function—especially the fault-aware and variability-aware regularization terms. This is important for understanding the contribution of each part of the framework.

Recommended Changes:

3. Discussion on Scalability to Larger Models:
   Add a brief section discussing how the method might scale to larger architectures such as Vision Transformers (ViTs) or LLMs, including expected challenges or adaptations needed.

4. Expand Related Work Comparison:
   Strengthen the comparison with recent post methods in quantization or robustness-aware training, particularly those targeting edge or neuromorphic deployment.

5. Optional Real-World Validation:
   While not mandatory, even limited testing on real hardware (e.g., an analog crossbar or neuromorphic chip) would significantly boost the practical credibility of the method.

**Strengths And Weaknesses:**

Strengths:

1. Unified and Flexible Framework:
   The paper presents a unified regularization-based framework that simultaneously addresses quantization, hardware faults, and device variability. This holistic approach is both technically sound and practically valuable for robust deployment on non-ideal edge hardware.

2. Strong Empirical Results Across Domains:
   The method achieves competitive or superior performance across a variety of tasks, including image classification with ANNs and event-based processing with SNNs. It shows notable robustness under extreme bit faults and variability levels, demonstrating strong generalization.

3. Hardware-Aware Design:
   The proposed quantization model is designed with hardware implementation in mind, supporting analog and digital crossbar architectures. The use of learnable bit multipliers maintains compatibility with fixed-point inference, ensuring practical applicability.

 Weaknesses:

1. No Support for Post-Training Quantization:
   The method requires training from scratch or fine-tuning with the regularization loss, which limits its use in scenarios where only pretrained models are available and retraining is not feasible.

2. Lack of Real Hardware Validation:
   All robustness evaluations are conducted through simulation. While the results are promising, experimental validation on real-world hardware platforms would strengthen the claims and demonstrate practical viability.

---

> ### Author Response · Authors · 2025-07-10
> **Response to Reviewer 91Ln**
>
> We thank the reviewer for their feedback and positive evaluation of our work. Below, we address each point raised and provide **additional experiments** to further substantiate our claims.
>
>
>     No Support for Post-Training Quantization: The method requires training from scratch or fine-tuning with the regularization loss, which limits its use in scenarios where only pretrained models are available and retraining is not feasible.
> This is certainly true for our 3-bit and 4-bit QAT experiments, which require training the network for the same number of epochs as the original training schedule for the floating-point baseline to achieve lossless quantization. However, given a quantized model, the fault-aware and variability-aware training requires only a fraction of the original training time. For our ImageNet experiments, this amounts to 10–20 epochs of fault-aware and variability-aware training, compared to 90–100 epochs of quantization-aware training needed to obtain optimal quantized weights. We have added a new **Section 5.5** to discuss the limitations and challenges of our method.
>
>     Lack of Real Hardware Validation: All robustness evaluations are conducted through simulation. While the results are promising, experimental validation on real-world hardware platforms would strengthen the claims and demonstrate practical viability.
> We acknowledge this criticism; however, it is unfortunately beyond our control at the moment. Hardware development timelines are typically quite long, and we are currently waiting for an in-group developed platform to mature to the required level for on-chip inference testing.
>
>     Clarify Limitations Regarding Post-Training Quantization: The paper should explicitly acknowledge that the method does not support post-training quantization and explain why this limitation exists. A brief discussion of potential future solutions would also be helpful.
> We have created a new **Section 5.5** to discuss the limitations and challenges of the method, specifically relating to post-training quantization and large transformer-based models.
>
>     Ablation Study on Loss Components: Include ablation experiments to isolate the effects of different components in the loss function—especially the fault-aware and variability-aware regularization terms. This is important for understanding the contribution of each part of the framework.
>
> We have performed a detailed ablation study on the impact of the different components of the fault-awareness strategy we have implemented, which is the most complex of the three methods developed in this work. To more clearly delineate the impact of various components, we have expanded the discussion in **Section 5.2** and updated **Fig. 7a** (previously Fig. 6a) and **Table 5** (previously Table 4).
>
> To summarize, we divide the different types of Fault-Aware Training into four distinct levels:
> - Basic awareness (i.e., periodic remapping of network weights based on fault maps, for example, every 4 epochs)
> - Passive mitigation (i.e., nearest valid level mapping [Hanif & Shafique, 2023], where errors due to bit-faults are minimized while mapping floating-point weights to fixed-point values)
> - Active mitigation (i.e., [Biswas & Ganguly, 2024], with regularization-based fault-aware training but without nearest valid level mapping)
> - Active + passive mitigation (this work, combining regularization-based fault-aware training with nearest valid level mapping).
>
> We perform fault-aware training on the ResNet-18 network using ImageNet with various bit-fault levels under all four configurations described above. As shown in the current **Fig. 7a** (previously Fig. 6a), the active + passive mitigation approach performs best at preserving network performance in the presence of bit-faults, followed by active mitigation, which outperforms passive mitigation. Basic awareness performs the worst.

---

> ### Author Response · Authors · 2025-07-10
> **Response to Reviewer 91Ln (2)**
>
> .
>
>     Discussion on Scalability to Larger Models: Add a brief section discussing how the method might scale to larger architectures such as Vision Transformers (ViTs) or LLMs, including expected challenges or adaptations needed.
> We have added a discussion on applying our method to transformer-based models in the new **Section 5.5**, under Limitations, Challenges, and Scope.
>
>     Expand Related Work Comparison: Strengthen the comparison with recent post methods in quantization or robustness-aware training, particularly those targeting edge or neuromorphic deployment.
> We have expanded our fault- and variability-aware training experiments in **Tables 5 and 6** (previously Tables 4 and 5, respectively) to more effectively compare our method against existing literature. In the new **Section 5.5**, we also discuss state-of-the-art quantization-based optimization techniques applied to transformer-based models, identifying promising directions for scaling our method to larger networks.

---

### Review · Reviewer_vxmp · 2025-06-11

**Summary Of Contributions:**

This paper proposed to use regularization-based method as a unified framework to handle quantization and fault/variation tolerance given a compute-in-memory accelerator scenario.  The author explored several quantization scheme, including typical linear, fixed step size and a more flexible "bit-multiplier" option, and then proposed that "bit-multiplier" provides better quantization outcome while being hardware-friendly for compute-in-memory applications.  Furthermore, the author added the stuck-at fault as a regularization term together with the quantization term to the loss function, which basically removes those "dead bits" from optimization. Note that the "dead bits" value will be replaced periodically by "nearest valid bit mapping" during the training.  Similarly, variation in low-resistance state is modeled through a "variability map" that modulates the ideal multiplier vector in a element-wise manner. The author observed significant improvement on fault and variation tolerance, especially under 4-bit high fault rate (20%) and/or high variation (40%) settings.

**Audience:**

Yes

**Claims And Evidence:**

No

**Requested Changes:**

plz see weakness above

**Strengths And Weaknesses:**

Strength
1. studied both DNN and SNN.


Weakness
1. Effectiveness of regularization-based bit-multiplier QAT method.
    - To evaluate a new quantization scheme, it would be crucial to investigate a variety of bit settings, networks, and datasets, in order to verify the generalization of the method. For example, popular PTQ papers BRECQ demonstrated their works on a series of classification CNNs, object detection nets, and transformers. Even the works published relatively early like LSQ+ paper has showed 3 different bit settings over 3 different networks. (By the way, LSQ+ Table 4 also outperformed Table 2 in this work.) It would not be a well-supported claim to state "When compared to other methods, out approach matches or outperforms existing techniques..."
    - Another possible confusion from Eq 6 and 7 is that readers would wonder that why the additional term only considers ||W-Wq|| instead of ||xW - xWq||, as typically used in most of the PTQ papers. Minimizing the quantization errors of W is not sufficient to guarantee minimized xW, as activation x is also quantized in this work.
    - CIFAR10 is just not representative enough for evaluating a quantization method, please consider moving Table 1 to Appendix.



2. Fault tolerance
    - Table 4 tried to make a comparison between different works. But network, dataset, and precision bit setting are all different, which makes it really hard to draw useful information. Maybe author could add a few items using the config of previous works, such as ResNet18/CIFAR10/8b, so that readers can get the idea easier.

    - It's a bit unclear where the improvement in Fig 6a was from. Section 3.6 states the nearest valid level mapping technique is the same as previous work (Hanif & Shafique 2023), which is assumed to be the red dashed line in Fig 6a. Blue dashed line in the same plot is derived from applying the faulty bits mask as a regularization term (based on Biswas & Ganguly 2024), whose effect is equivalent to excluding the faulty bits from optimization.  Assuming faulty bits' values for both approaches are updated every 4 epoch (based on nearest levels), and assuming in Hanif's method, between updates the faulty bits will be "optimized" as opposed to frozen in Biswas's approach, the main difference seems to be whether to freeze the faulty bits between updates. But the significant accuracy dropped, i.e. from ~65% to ~15% at 20% stuck-at rate, almost seems like the training was not completed or the convergence is poor. Since this is one of the main claims of this paper, it would be better to elaborate a little more in Section 4.3/5.2. Currently, these paragraphs only provide observations. Some discussion of possible causes or even some ablation experiments would make this paper much stronger.

---

> ### Author Response · Authors · 2025-07-10
> **Response to reviewer vxmp**
>
> We thank the reviewer for the constructive feedback. Below, we address each point in detail and include **additional experiments** to further support our claims.
>
>     To evaluate a new quantization scheme, it would be crucial to investigate a variety of bit settings, networks, and datasets, in order to verify the generalization of the method. For example, popular PTQ papers BRECQ demonstrated their works on a series of classification CNNs, object detection nets, and transformers. Even the works published relatively early like LSQ+ paper has showed 3 different bit settings over 3 different networks. (By the way, LSQ+ Table 4 also outperformed Table 2 in this work.) It would not be a well-supported claim to state "When compared to other methods, out approach matches or outperforms existing techniques..."
>
> We have updated **Table 2** with 3-bit QAT results for the ResNet-18 network trained on ImageNet. Additionally, we have added a new **Table 3** containing 4-bit and 3-bit QAT results for the ResNet-50 network, also trained on ImageNet. Furthermore, we have improved our QAT method to include quantization in the forward pass with a Straight-Through Estimator (STE) in the backward pass, as described in Section 3.3. This improvement has led to better QAT results for ResNet-18, as shown in **Table 2**.
>
> With these latest updates, we now present QAT results for multiple networks and two different bit settings, in addition to our QAT results on Spiking Neural Networks. Finally, we would like to mention that we have focused on classification problems in this paper, as the all-or-nothing nature of classification accuracy (especially top-1) makes it highly sensitive to small quantization errors.
>
>     Another possible confusion from Eq 6 and 7 is that readers would wonder that why the additional term only considers ||W-Wq|| instead of ||xW - xWq||, as typically used in most of the PTQ papers. Minimizing the quantization errors of W is not sufficient to guarantee minimized xW, as activation x is also quantized in this work.
> It is indeed common practice in Post-Training Quantization (PTQ) papers to minimize the error in the pre-activations (‖xW - xqWq‖) over a small calibration set of samples. This approach is well-suited for obtaining optimal quantization parameters given a fixed calibration set.
>
> However, in this paper, we propose a Quantization-Aware Training (QAT) method, where the quantization parameters for both weights and activations are typically optimized independently, alongside the optimization of the weights themselves. This provides greater flexibility for the network and learning algorithm to explore the parameter space in search of optimal solutions, both for weights and quantization parameters, at the cost of increased training and optimization time.

---

> ### Author Response · Authors · 2025-07-10
> **Response to Reviewer vxmp (2)**
>
> .
>
>     Table 4 tried to make a comparison between different works. But network, dataset, and precision bit setting are all different, which makes it really hard to draw useful information. Maybe author could add a few items using the config of previous works, such as ResNet18/CIFAR10/8b, so that readers can get the idea easier.
>
> We have addressed a similar criticism in detail in response to the fourth point raised by **Reviewer nvNQ**. Please see the relevant response above.
>
>     It's a bit unclear where the improvement in Fig 6a was from. Section 3.6 states the nearest valid level mapping technique is the same as previous work (Hanif & Shafique 2023), which is assumed to be the red dashed line in Fig 6a. Blue dashed line in the same plot is derived from applying the faulty bits mask as a regularization term (based on Biswas & Ganguly 2024), whose effect is equivalent to excluding the faulty bits from optimization. Assuming faulty bits' values for both approaches are updated every 4 epoch (based on nearest levels), and assuming in Hanif's method, between updates the faulty bits will be "optimized" as opposed to frozen in Biswas's approach, the main difference seems to be whether to freeze the faulty bits between updates. But the significant accuracy dropped, i.e. from ~65% to ~15% at 20% stuck-at rate, almost seems like the training was not completed or the convergence is poor. Since this is one of the main claims of this paper, it would be better to elaborate a little more in Section 4.3/5.2. Currently, these paragraphs only provide observations. Some discussion of possible causes or even some ablation experiments would make this paper much stronger.
>
> We would like to thank the reviewer for this pointed and helpful criticism. To more clearly delineate the impact of various components, we have expanded the discussion in **Section 5.2** and updated **Fig. 7a** (previously Fig. 6a) and **Table 5** (previously Table 4).
>
> To summarize, we divide the different types of Fault-Aware Training into four distinct levels:
> - Basic awareness (i.e., periodic remapping of network weights based on fault maps, for example, every 4 epochs)
> - Passive mitigation (i.e., nearest valid level mapping [Hanif & Shafique, 2023], where errors due to bit-faults are minimized while mapping floating-point weights to fixed-point values)
> - Active mitigation (i.e., [Biswas & Ganguly, 2024], with regularization-based fault-aware training but without nearest valid level mapping)
> - Active + passive mitigation (this work, combining regularization-based fault-aware training with nearest valid level mapping).
>
> We perform fault-aware training on the ResNet-18 network using ImageNet with various bit-fault levels under all four configurations described above. As shown in the current **Fig. 7a** (previously Fig. 6a), the active + passive mitigation approach performs best at preserving network performance in the presence of bit-faults, followed by active mitigation, which outperforms passive mitigation. Basic awareness performs the worst.

---

### Review · Reviewer_nvNQ · 2025-06-26

**Summary Of Contributions:**

This paper proposes a quantization-aware training (QAT) framework to the non-ideal behavior of hardwares by introducing a novel regularization term.
This method enables fault- and variability-aware fine-tuning, demonstrating substantial improvements under severe bit-fault rates and device variability in some CNN and SNN architectures.

**Audience:**

Yes

**Claims And Evidence:**

Yes

**Requested Changes:**

**Requested Changes**
- Clearly explain the hardware-aware optimization workflow. Is this optimization (including fault analysis and retraining) performed on the faulty hardware, or is it performed on the GPU after the faulty parts have been separately identified?

- Improve mathematical notation clarity.

- Include results for ResNet-50 or similarly sized networks to demonstrate the scalability of the proposed method.

- While noted as future work, even a limited experiment on a tiny transformer model would improve the paper. So, add the evaluation of transformer architecture if possible.

- Rewrite or supplement comparisons in Tables 4 and 5. Since model architectures and datasets differ, conduct experiments under the same conditions as existing methods.

- Please address the content in the weakness section as much as possible.

**Questions**

- I am not sure why FP (floating-point) baselines in Tables 1 and 2 show significant variation across methods. If there is a reason for this, could you please explain it to me?

**Strengths And Weaknesses:**

**Strengths**
- This paper addresses hardware-level defects that occur in the implementation of edge AI hardware from the perspective of the loss function. This approach is interesting.

- The proposed method shows benefits in robustness against hardware faults, particularly under high fault/variability conditions.

**Weaknesses**
- The optimization pipeline lacks sufficient clarity: it's unclear where and how optimization occurs (on-GPU or on-chip), and how hardware faults are identified and fed back into the training process.

- Some mathematical notations are confusing or insufficiently explained. e.g.,
  - $\langle \cdot \rangle$
  - $W_{level} \rightarrow W_{\textrm{level}}$
  - $var \rightarrow \textrm{var}$

- Evaluation is limited to small-scale models. The scalability of the proposed method to larger models (e.g., ResNet-50) remains uncertain.

- Evaluation tables (especially Tables 4 and 5) compare across different models, datasets, and hardware conditions, making it difficult to conclude that the proposed method is superior to existing methods.

- The paper lists experiments with Transformer-based models as future work. However, given that Transformers are currently one of the most widely used architectures, I think the lack of such experiments is one of the weakness of this paper.

---

> ### Author Response · Authors · 2025-07-10
> **Response to Reviewer nvNQ**
>
> We appreciate the thoughtful feedback and insightful questions. Below, we address each point and provide **additional experiments** to further substantiate our claims.
>
>      The optimization pipeline lacks sufficient clarity: it's unclear where and how optimization occurs (on-GPU or on-chip), and how hardware faults are identified and fed back into the training process.
> In our method, as in most of the related literature we have compared against (e.g., [Peng et al., 2020], [Hanif & Shafique, 2023]), the optimization is performed offline using a software emulation of the characterized on-chip hardware effects. Device-aware training is typically conducted on standard GPU-based training setups.
>
> An exception to this is chip-in-loop optimization [Gonugondla et al., 2018], where the forward pass is executed on the chip, followed by offline backpropagation. Weight updates are computed using the on-chip activations, and the updated weights are then written back to the chip.
>
> This difference in approach is illustrated in **Figure 3** (newly added) on page 5.
>
>     Some mathematical notations are confusing or insufficiently explained.
> We apologize for the inconsistent and unexplained notation. We have cleaned up the inconsistencies in the mathematical notation and included the definition of the ⟨a, b⟩ operation (which denotes the inner product or matrix multiplication) in **Section 3.2**.
>
>     Evaluation is limited to small-scale models. The scalability of the proposed method to larger models (e.g., ResNet-50) remains uncertain.
>
> We have added relevant ResNet-50 experiments in **Table 3 (newly added), Table 5 (previously Table 4), and Table 6 (previously Table 5)**. Furthermore, we have expanded our Quantization-Aware Training (QAT) experiments on the ImageNet benchmark to include 3-bit weight and activation quantization, and have updated Tables 2 and 3 accordingly.
>
> Finally, we have improved our QAT method to include quantization in the forward pass with a Straight-Through Estimator (STE) in the backward pass, which has resulted in improved QAT performance for ResNet-18. Details of the updated QAT method are provided in Section 3.3.
>
>
>     Evaluation tables (especially Tables 4 and 5) compare across different models, datasets, and hardware conditions, making it difficult to conclude that the proposed method is superior to existing methods.
>
> **Table 5 (previously Table 4)**
>
> - We have performed Fault-Aware Training (FAT) on the ResNet-18 network using the CIFAR-10 dataset with 10% and 20% bit fault rates and 3-bit quantization, for direct comparison with [Hanif & Shafique, 2023]. Our method demonstrates better robustness to stuck-at faults at lower bit quantization.
>
> - We have performed Fault-Aware Training on the ResNet-50 network using ImageNet with 5% and 10% bit fault rates and 4-bit quantization, for a meaningful comparison with [Koppula et al., 2019]. The DenseNet201 network, for which [Koppula et al., 2019] reported Fault-Aware Training results on ImageNet, is approximately equivalent to ResNet-50 in terms of both the number of parameters (~20 million and ~26 million, respectively) and ImageNet top-1 performance. Once again, our method shows improved results at comparatively lower bit quantization.
>
> - We have not specifically performed architecture-matched fault-aware training experiments to compare against [Biswas & Ganguly, 2024] (10-layer CNN on CIFAR-10 with a 10% bit fault rate and 3-bit quantization) and [Koppula et al., 2019] (ResNet-101 on CIFAR-10 with a 4% bit fault rate and 8-bit quantization), as we have demonstrated improved fault-aware training results on CIFAR-10 using equivalent or smaller networks (VGG13 and ResNet-18, respectively) at 3-bit quantization and a higher bit fault rate (20%).
>
> - Finally, we have removed the MNIST result rows to streamline the table and focus on more relevant benchmarks and standard network architectures.
>
> **Table 6 (previously Table 5)**
>
> - We have performed Variability-Aware Training (VAT) on the ResNet-50 network using ImageNet with 20% variability and 4-bit and 6-bit quantization, for direct comparison with [Han et al., 2024]. Our method demonstrates better adaptation to higher variability at lower bit quantization.
>
> - We have performed Variability-Aware Training on the VGG-13 network using CIFAR-10 with 20% variability and 4-bit quantization, for a meaningful comparison with [Peng et al., 2020].

---

> > ### Author Response · Authors · 2025-07-10
> > **Response to Reviewer nvNQ (2)**
> >
> > .
> >
> >     Clearly explain the hardware-aware optimization workflow. Is this optimization (including fault analysis and retraining) performed on the faulty hardware, or is it performed on the GPU after the faulty parts have been separately identified?
> > We have added a new **Figure 3** to illustrate our hardware-aware optimization workflow and to compare it with the alternative chip-in-loop optimization approach.
> >
> >     Improve mathematical notation clarity.
> > We have cleaned up notational inconsistencies and added explanations for previously unexplained notations in **Section 3.2**.
> >
> >     Include results for ResNet-50 or similarly sized networks to demonstrate the scalability of the proposed method.
> > We have added relevant ResNet-50 results for QAT, FAT, and VAT in **Tables 3 (newly added), 5 (previously Table 4), and 6 (previously Table 5)**, respectively.
> >
> >     While noted as future work, even a limited experiment on a tiny transformer model would improve the paper. So, add the evaluation of transformer architecture if possible.
> > While there are no architectural barriers to adapting our method to transformers, the large training requirements make it necessary to shift from QAT toward PTQ. We are currently exploring how to incorporate this into our method, which is why we have not yet developed a relevant evaluation in the transformer space. This is discussed in more detail in the newly added **Section 5.5**.
> >
> >     Rewrite or supplement comparisons in Tables 4 and 5. Since model architectures and datasets differ, conduct experiments under the same conditions as existing methods.
> > We have augmented **Tables 5 and 6** (previously Tables 4 and 5, respectively) with architecture- and benchmark-matched experiments at lower bit quantization and at higher or equivalent fault/variability levels.
> >
> >     I am not sure why FP (floating-point) baselines in Tables 1 and 2 show significant variation across methods. If there is a reason for this, could you please explain it to me?
> > The floating-point (FP) baselines show some variation depending on the choice of hyperparameters such as learning rate, optimizer, learning rate schedule, and number of training epochs. Since this affects the performance of QAT, we also report the degradation (or improvement, in some cases) relative to the floating-point baseline as the primary value for comparison.

---

### Author Response · Authors · 2025-07-10
**Response to reviewers (General)**

We thank the reviewers for their insightful comments and constructive suggestions. To the best of our ability, we have incorporated the review comments to improve our work, and we believe this has enhanced the quality of the paper.

Below, we provide a point-by-point response to the review comments, along with a record of the changes made to address the suggestions. We have conducted **multiple additional experiments** to validate the claims made in our paper. The changes in the manuscript are highlighted in yellow.

---

### Decision · Action_Editor_JTnN · 2025-09-04

**Recommendation:** Reject

**Audience:**

No

**Audience Explanation:**

According to the review comments, the current formulation and experimental results cannot well support the advantage of the proposal.

**Claims And Evidence:**

No

**Claims Explanation:**

The paper presents a flexible QAT framework that addresses quantization, hardware faults, and variability. Reviewers present several concerns regarding the proposal, including the hardware efficiency of the proposed n-multiplier quantization scheme, limited experimental analysis, fairness of the empirical comparison. During the rebuttal period, the authors failed to address any of them properly. Given these problems I recommend rejection.

**Resubmission Of Major Revision:**

The authors may consider submitting a major revision at a later time.

---

> ### Author Response · Authors · 2025-09-09
>
> Dear Action Editor,
>
> We are struggling to reconcile the abrupt turnaround on the evaluations (Claims and Evidence/Audience) between the first reviews and this decision. Moreover, the response is lacking in any details that could help us identify directions of improvement in the major revision. Please provide more detailed feedback.
>
> Thank You